# The Effect of Acetylation on the Physicochemical Properties of Chickpea Starch

**DOI:** 10.3390/foods12132462

**Published:** 2023-06-23

**Authors:** Chunlan Zhang, Mengyao Du, Tiantian Cao, Wei Xu

**Affiliations:** 1College of Food Science and Engineering, Tarim University, Alar 843300, China; 2Production & Construction Group Key Laboratory of Special Agricultural Products Further Processing in Southern Xinjiang, Alar 843300, China; 3College of Life Science, Xinyang Normal University, Xinyang 464000, China

**Keywords:** chickpea starch, acetylation treatment, physicochemical properties

## Abstract

The effect of acetylation on the physicochemical properties of chickpea starch was studied. After the chickpea starch was acetylated, the basic properties were measured. When the degree of substitution (DS) was 0.1004 and the temperature was 95 °C, the solubility and swelling power of starch were 19.6% and 21.4 g/g, respectively. The freeze–thaw stability of acetylated starch paste increased with the increase in the degree of substitution. The surface morphology of starch granules changed, but the crystalline morphology did not change, and the C-type crystalline structure was still maintained. There are three new absorption peaks in the infrared spectroscopy of starch, and the -COCH_3_ group was introduced. With the increase in DS, the viscosity of esterified chickpea starch decreased gradually. Compared with unmodified chickpea starch, the ability to form gel was poor.

## 1. Introduction

*Chickpeas* are one of the main bean crops in the world. They are named for their beak-shaped protuberance near the navel. It is also known as *peach bean* and *chicken pea*. It is widely planted in the west of our country and is very resistant to drought and infertility; it has great market potential [1]. Chickpeas are rich in starch, protein, dietary fiber, vitamins, and other nutrients, which have great benefits for human health [2]. Among them, carbohydrates and protein account for about 80% of the dry weight of seeds, especially starch contents of 40–60%, so chickpeas are a good source of starch.

Starch is a kind of natural macromolecular compound that is abundant in nature and has the advantage of a low price. It is easy to obtain, renewable, degradable, and non-polluting to the environment [3,4]. Therefore, for a long time, countries all over the world have attached great importance to the development and utilization of starch resources [5]. But the nature of native starch (NS) is not ideal, usually there is thermal decomposition, low solubility, and other defects [6]. For example, the gelatinization stability of native starch is poor, easy to dehydrate in the freezing process, and the freeze–thaw stability is not good [7]. It was not suitable for some processing types. Starch undergoes some chemical modifications, such as acetylation [8], carboxymethylation [9], oxidation [10], and hydroxypropylation [11], to satisfy different food needs [12]. In the food industry, especially in the case of native starch, which could not provide the best performance, the introduction of new groups will change the structure, which can improve the functional performance of starch after chemical modifications of the starch structure.

Acetylation is a kind of chemical modification. The starch was chemically modified with acetic anhydride, using NaOH as the catalyst. The hydroxyl group of glucose in starch was replaced by the acetyl group, which results in some changes in the starch’s molecular structure. The main purpose of acetylated-modified starch is to change the properties of starch, such as the physicochemical properties, gelatinization, gel texture, crystal morphology, etc. The gelatinization properties and aging of starch modified by acetylation are very different from that of native starch. Lin et al. found that, compared with native starch, acetylated starch had better solubility and swelling powers [13]. Colussi et al. showed that the gelatinization temperature of starch paste after acetylation was significantly lower, and the gelatinization temperature gradually decreased with the increase in the acetyl content [14].

The functional properties of acetylated starch changed with the degree of substitution. The main functional properties of acetylated starch depend on the type of starch, degree of substitution, and size of the acetyl group; the content of the acetyl group and DS in acetylated starch depend on the reaction time, catalyst, plant source, and amount of reagent [15]. Some starches have a low degree of substitution of about 0.01–0.2 and can be used as adhesives, thickeners, plasticizers, film-forming agents, stabilizers, and adhesives. They are often used as additives in the food industry [16,17], such as baked goods, frozen foods, snack foods, and so on [18]. Acetylated starch also has a medium degree of substitution of 0.2–1.5, a high degree of substitution of 1.5–3.0, and is highly soluble in organic solvents, and can therefore be used as a thermoplastic material. In this paper, low DS-acetylated chickpea starch with different DS was prepared, and its physicochemical and functional properties were studied to improve the functional properties of the starch. This study will promote the development and utilization of chickpeas provide a theoretical basis for chickpeas.

## 2. Materials and Methods

### 2.1. Materials

Chickpea starch was prepared by dilute alkali method in laboratory; the starch content was 95 DW%. Acetic anhydride was purchased from Comeo Chemical Reagent Co., Ltd. (Tianjin, China). Sodium hydroxide and potassium bromide were purchased from Chemical Reagents Co., Ltd. (Nanjing, China). Hydrochloric acid was purchased from Jing en Industrial Co., Ltd. (Shanghai, China).

### 2.2. Extraction of Chickpea Starch

The chickpeas were ground and passed through 80 mesh; then, 5 g of the chickpea powder and NaOH solution (0.3% concentration) were mixed at ratio 1:4 of material to liquid. Under the condition of 40 °C, the mixture was stirred at constant temperature for 240 min in a magnetic stirring water bath and centrifuged at 4000 r/min after stirring. After centrifugation, the top layer of black protein solution was scraped off, the bottom layer of starch was washed with water, the starch was centrifuged until it was pure white, dried in a 40 °C drying box for 24 h, and then crushed using 80-mesh sieve.

### 2.3. Acetylation of Chickpea Starch

Acetylated-modified starch was prepared with acetic anhydride as reagent. A 40% starch slurry was prepared by mixing chickpea starch with water. The range of acidity of the starch slurry was pH 8.0–8.5. The acetic anhydride *v*/*w* (4, 6, 8, and 10%) was added slowly to keep the pH in the range of 8.0–8.5. After adding acetic anhydride within 20 min and continuing the reaction for 1 h, the reaction was stopped by adjusting the pH to 6.5 with 0.5 mol/L HCL, the starch was washed to neutral, then dried in a 40 °C oven for 24 h, and crushed to obtain acetylated starch [19].

### 2.4. Measurement of Degree of Substitution

A total of 2 g of acetylated starch was put into iodine flask, and 50 mL of distilled water was added. Phenolphthalein was used as indicator; 0.1 mol/L NaOH was added to make the solution turn reddish, and 10 mL of 0.5 mol/L NaOH was added. After shaking for 0.5 h on the oscillator, the sample was titrated with 0.5 mol/L HCL until the color disappeared, and the volume of consumed HCL was recorded as V_1_. Blank: using chickpea unmodified starch, consumption HCL volume was recorded as V_0_. According to Formula (1), which concerns the acetyl content of A [20].
(1)$$A=\frac{\left(V_{1}-V_{0}\right)C\times 0.043}{m}$$

In the formula, A is acetyl content, %; V_0_ is blank consumption HCL volume, mL; V_1_ is consumption HCL volume, mL; C is HCL concentration, and mol/L; m is the quality of the starch, g.

The degree of substitution DS is calculated in Formula (2):(2)$$\mathrm{DS}=\frac{162A}{\left(4300-42A\right)}$$

In formula, DS is degree of substitution, A is acetyl content, %.

### 2.5. Determination of Basic Properties

#### 2.5.1. Determination of Solubility and Swelling Power

The starch sample M was taken and prepared as a 1% starch suspension. It was stirred in magnetic stirring water bath (55, 65, 75, 85, and 95 °C) for 30 min and cooled at room temperature. The supernatant was poured into a weighing bottle after it was centrifuged at 4000 r/min for 10 min. Then, the supernatant was dried in 100 °C oven to constant weight and weighed (label as A). The solubility (S), the precipitate (P), and swelling power (B) of starch were calculated using Formulas (3) and (4) [21].
(3)$$S\left(\%\right)=\frac{A}{M}\times 100\%$$
(4)$$B\left(g/g\right)=\frac{P}{M\times (1-S)}\times 100$$

#### 2.5.2. Measurement of Transparency

To prepare 1% starch suspension, it was stirred and heated at 100 °C for 0.5 h. Then, the paste was cooled at room temperature. The distilled water was used as a blank control. The absorbance A was determined at 620 nm with photometer, and the transparency of starch was calculated. Formula (5) is as follows [22]:(5)$$T=\frac{1}{10^{A}}$$

#### 2.5.3. Determination of Freeze–Thaw Stability of Starch Paste

A total of 6% starch suspension was prepared, after it was treated at 100 °C for 30 min, stored at 4 °C for 24 h, and then put into a centrifuge tube of known weight (m_1_). Then, the sample was stored at −18 °C for 24 h. It was centrifuged at 3000 r/min for 20 min after thawing. The sediment was weighed and labeled m_2_. To calculate the precipitation rate according to Formula (6), the above steps were cycled five times [23].
(6)$$I=\frac{\left(m_{1}-m_{2}\right)}{m_{1}}\times 100\%$$

### 2.6. Structure Determination

#### 2.6.1. Observation of Starch Granule Morphology

The granule morphology of the sample was analyzed by scanning electron microscope (Apero S, Lecht instruments and Equipment Co., Ltd. Shenzhen, China). The samples were evenly dispersed on the conductive adhesive, and then placed on the coating platform for gold spray treatment. The granule morphology was observed at an accelerated voltage of 5.0 KV, and micrographs were taken at 3000× magnification [24].

#### 2.6.2. X-ray Diffraction (XRD)

The sample was dried and pressed. The samples were analyzed by X-ray diffraction (LP-XRD, Kirshida Electronic Technology Co., Ltd. Shenzhen, China). The diffraction conditions were as follows: copper target voltage of 40 kV, current of 40 mA, and CuKa radiation. The scanning range was 5° to 40°, the step length was 0.02°, and the scanning speed was 2°/min [25].

#### 2.6.3. Fourier Transform Infrared (FT-IR) Spectroscopy

The dried starch sample was mixed evenly with the KBr powder and quickly ground for 1 min, and then press the tablet. The thin film was scanned with an FTIR spectrophotometer (LP-FTIR-300, Shimadzu Co., Ltd., Kyoto, Japan). The spectral resolution was 4 cm^−1^; the scanning range was 4000 cm^−1^ to 400 cm^−1^; and the number of scans was 32 [26].

#### 2.6.4. Determination of Starch Gelatinization Curve (RVA)

The properties of starch gelatinization were determined by rapid viscosity analyzer (RVA0524, Huishi Instrument Equipment Co., Ltd., Shanghai, China). The 3.0 g starch sample and 25 mL of water were mixed. The starch slurry was equilibrated at 50 °C for 1 min. The measuring temperature ranged from 50 °C to 95 °C, and the heating rate was 12 °C/min. The temperature was held at 95 °C for 2.7 min, and the slurry was reduced to 50 °C for 1 min. We recorded peak viscosity, trough viscosity, breakdown viscosity, final viscosity, setback viscosities, and gelatinization temperature [27].

#### 2.6.5. Determination of Gel Texture of Starch

The concentration of starch suspension was 12%, and it was heated in a boiling water bath. Then, the cooled sample was placed at 4 °C for 24 h. To measure the gel properties of the samples, the TPA compression mode of texture analyzer was used. The pre-test speed was 2 mm/s; the mid-test speed was 5 mm/s; and the post-test speed was 5 mm/s. The compression rate was 50%. The probe was P/0.5 R [28].

### 2.7. Data Analysis

Significance analysis was performed using SPSS Statistics 25 one-way ANOVA test and Duncan, and relative crystallinity was calculated using Jade software.

## 3. Results and Analysis

### 3.1. Acetyl Content and Degree of Substitution of Chickpea Starch

The effect of acetic anhydride on the acetyl content and the degree of substitution of chickpea starch is shown in Table 1. The highest acetyl content and degree of substitution of acetylated starch were 2.60 ± 0.02% and 0.1004 ± 0.001 when the acetic anhydride content was 10%. This indicates that when the acetyl content and DS increased with the increase in the acetic anhydride content, there was significant difference (*p* < 0.05). The four acetylated-modified starches were all low DS starches.

### 3.2. Basic Property Analysis

#### 3.2.1. Changes in the Solubility and Swelling Power

As shown in Figure 1 and Figure 2, the solubility and swelling power of native chickpea starch and modified starches with four different DS were significantly different (*p* < 0.05). The greater the degree of substitution, the greater the solubility and swelling power of the starch. When the starch was heated in water, the hydrogen bond of the stable double helix structure was broken, the hydrogen bond was replaced by the water molecule, and the starch granule expanded and increased in size. Compared with the native starch, the acetylated starch retained more water, and the solubility of starch treated with acetic anhydride was higher at the same heating temperature. When DS was 0.1004 and the temperature was 95 °C, the solubility and swelling power of starch were 19.6% and 21.4 g/g, respectively, which were 4.4% and 4.8 g/g higher than those of native starch at 95 °C. The addition of an acetyl group to starch breaks the hydrogen bond between starch and water, changes its internal structure, and increases its solubility. Compared with native starches, the swelling degree of acetylated chickpea starches increased, which may be due to the following two main reasons: acetylated starches introduced acetyl groups, which made the molecular distance between chickpea starches become larger, and the steric hindrance of chickpea starches increased, which is unfavorable to the recovery of starch cohesion in chickpeas [29]. The hydrophilicity of the acetyl group weakens the hydrogen bonding force between chickpea starch molecules, and a large number of water molecules enter the starch granules, resulting in the swelling of the starch granules and a large number of water molecules permeating the starch granules, making it more likely to be near the amorphous area of the starch and thus causing the starch granules to expand in a limited range. It was found that the solubility and swelling power of starches with different degrees of substitution increased in varying degrees [30], and when the DS increased, the solubility and swelling power also increased, indicating that the greater the DS, the better the solubility and swelling power of starch.

#### 3.2.2. Changes in Transparency

The transparency of native chickpea starch was significantly different from that of acetylated starch (Figure 3). As can be seen from the diagram, the DS is greater, the transparency is greater, and the transparency of starch paste after acetylation is higher than that of natural starch. When DS was 0.1004, the maximum transparency of starch was 20.3%, which is 10.13% higher than that of natural starch. Many factors will affect the transparency of starch paste. Since acetyl is a hydrophilic group, a large number of acetyl groups entered the amorphous region of starch, which weakened hydrogen–bond interactions. Water molecules entered the granules, leading to an increase in the water content and the formation of the highly expanded starch; thus, the transmission of light improved. At the same time, the molecular structure of starch changed. The acetyl group replaced the original hydroxyl group, which increases the steric hindrance in the starch. The chains repel each other, and starch binds more easily to water, resulting in increased transparency [31]. Another important factor is that the starch granules are heated with water to form a starch paste. The transparency is closely related to the dissolution degree of the starch granules. When chickpea starch is dissolved in water, starches with high solubility have a higher degree of transparency. Acetylation causes the starch to become highly inflated, absorbing large amounts of water molecules, which weakens the starch particles’ ability to reflect. This allows more light to pass through the starch particles, thereby increasing the transparency of the starch paste. The addition of the acetyl group caused repulsion between the neighboring molecules of chickpea starch, and the hydrogen bond was greatly weakened.

#### 3.2.3. Changes in the Syneresis Rate

The syneresis rate is often used as an indicator of the freeze–thaw stability of starch paste. Generally speaking, a lower syneresis rate means better freeze–thaw stability. As shown in Figure 4, the freeze–thaw stability of starch decreased when the number of freeze–thaw increased. After two incidences of freeze–thaw, chickpea starch paste begins to appear spongy and completely loses its initial gelatinous state. The syneresis rate of native chickpea starch reached 46.3% when it finished five freeze–thawing. The syneresis rates of modified starch with different DS were 41.5%, 38.9%, 37.6%, and 35.2%, respectively, after the fifth freeze–thawing. The syneresis rate of native chickpea starch was significantly higher, and the syneresis rate of acetylated starch decreased with the increase in DS. This indicates that the water-holding capacity of chickpea starch paste improves after esterification; therefore, the freeze–thaw stability improves effectively. The reduction in dehydration shrinkage after modifications may be attributed to the entry of acetyl groups that interrupt interchain binding between starch chains. The starch molecules may rebind during freezing, but this binding is very limited. The hydrophilicity of the acetyl group increased the steric hindrance, so shrinkage was reduced [32]. In addition, because of steric hindrance of the acetyl group, amylose cannot be recombined during freezing and thawing, which increases the water-holding capacity of starch.

### 3.3. Structural Analysis

#### 3.3.1. The Change in Starch Granule Morphology

The scanning electron microscope morphology of acetylated starches with different DS was observed by SEM (Figure 5). The surface of starch granules in chickpea modified by acetylation was partly damaged, and grooves appeared. This indicated that the esterification reaction caused some damage to the surface of granules, but the appearance (size and shape) of starch is similar. The shape of starch is mostly kidney or oval-shaped. The surface of the acetylated starch has some grooves, but most of the particles are smooth and intact, without any corrosion or damage. As you can see from the diagram, the damage to the surface became more severe with the increase in DS, but the internal structure of starch remained intact. This indicates that acetylated starch only acts on the surface of granules. Some aggregation of granules can be found in Figure 5e. It may be that the damage to starch granules become more serious with the increase in the acetylation degree and the penetration by the acetyl group into the inner of the granule, and destroying the inner binding of the granule. Thus, the neighboring starch granules combine and cause the aggregation or fusion of the starch granule [33].

#### 3.3.2. Analysis of X-ray Diffraction

The XRD patterns of chickpea starch are shown in Figure 6. The absorption peaks of chickpea starch were found at 5.6°, 15°, 17 °, 18°, and 23°, respectively. The crystalline form of chickpea starch did not change obviously after acetylation [34]. The crystallinity of chickpea starch is shown in Table 2. The crystallinity of acetylated chickpea starch experienced no significant change, which may be due to the fact that a low DS cannot destroy the crystal structure of starch. The intensity and location of diffraction peaks of native chickpea starch and four different DS-acetylated starches were almost identical. This proved that the acetyl group had no effect on the crystal structure of starch.

#### 3.3.3. Analysis of the Fourier Transform Infrared Spectrum

In order to study the structure of acetylated starch, FTIR spectroscopy was used. The spectra of native chickpea starch, DS = 0.0486, 0.071, 0.0867, and 0.1004, are shown in Figure 7. The broad peak at 3400 cm^−1^ is the peak of –OH formed by bond breaking; the band at 2929 cm^−1^ is the stretching vibration peak of –CH–; the change in amylose and amylopectin contents result in the strength change in the –CH– expansion range [35]. Due to the tensile vibrations of the entire dehydrated glucose ring, several additional characteristic absorption bands appear at 929 cm^−1^, 857 cm^−1^, 763 cm^−1^, and 576 cm^−1^. Compared with those native chickpea starches, the acetylated starch showed three new absorption peaks due to the introduction of an acetyl group: the absorption peak at 1730 cm^−1^ was C=O, 1375 cm^−1^ was –CH_3_, and 1245 cm^−1^ was C-O [36]. This indicates that esterification occurred during acetylation and the –COCH_3_ group was introduced. The intensity of the absorption peak at 3400 cm^−1^ gradually decreased with an increasing DS compared with the native starch, which is mainly because of the continuous substitution of the –OH group by the introduction of –COCH_3_; so, the intensity at 3400 cm^−1^ of the original –OH absorption peak decreased gradually.

#### 3.3.4. Changes in the Viscosity Properties of Starch Paste

As shown in Figure 8, the RVA curve of starch shows that all the viscosity values of the acetylated chickpea starch have a decreasing trend. The RVA viscosity curve becomes smooth, which shows that acetylated starch paste has better stability of paste viscosity, and it is suitable as a thickening agent in food additives. The paste viscosity parameters of acetylated starch and native starch were significantly different (*p* < 0.05). When DS was 0.1004, the peak viscosity decreased by 989 cP, trough viscosity decreased by 660 cP, breakdown viscosity decreased by 329 cP, final viscosity decreased by 2649 cP, and the setback viscosity decreased by 1989 cP (Table 3). The viscosity of starch is determined by its relative molecular weight, the structure of the starch, and the size of the chain. Firstly, the hydrogen–bond interaction is weakened by the entry of the acetyl group, and the glycoside bond is broken by esterification, which will form more short amyloses, leading to a decrease in viscosity. Secondly, the structure of amylopectin in starch is destroyed, the intermolecular force of starch is weakened, and the inherent double helix structure of starch is changed to a single helix structure, which leads to a decrease in the viscosity and a decrease in the breakdown viscosity. Therefore, starch has a higher thermal paste viscosity stability. When the setback viscosity decreases, it indicates that the retrogradation of starch is weaker. This may be because acetyl blocks the rearrangement of starch molecules. There was a decrease in gelatinization temperature, which was due to the acetyl group entering the non-crystalline region, which destroyed the original stability of the starch granules and broke the intrinsic network structure; therefore, it is more advantageous for small water molecules to penetrate the starch granules. Finally, it decreased the starch gelatinization temperature [37,38]. The results indicate that after acetylation, the viscosity and gelatinization temperature decreased.

#### 3.3.5. Changes in the Gel Properties of Starch

As shown in Table 4, there have been great changes between the gel properties of native chickpea starch and acetylated starch. The gel properties of four different DS-acetylated starches were different from those of natural chickpea starch (*p* < 0.05). The gel parameters of acetylated starch were lower than those of native chickpea starch. When acetylated starch DS was 0.1004, the hardness decreased by 19.65 g, the cohesion decreased by 0.22 g, the elasticity decreased 1.5 mm, and adhesion and chewiness decreased by 16.33 g·s and 79.2 mj, respectively. The results indicate that the starch modified by acetylation was not easily retrograded and had better gel properties. Zdybel E. [39] argues that the strength of the gel depends on the binding force within the particles. The strength of starch gel is closely related to the content of amylose in starch. Because of the introduction of the acetyl substituent group, the binding force in these granules is hindered, and the hydrogen bond is weakened. Therefore, the formation of a stable gel network was hindered, resulting in weak gel strength in esterified starch.

## 4. Conclusions

In this paper, chickpea starch was acetylated. The FTIR spectra showed that acetylation introduced the COCH_3_ group. There were no changes in the crystal structure of starch by XRD. The viscosity of acetylated starch decreased with the increase in DS, and the ability to form gel decreased. However, acetylated starch has a higher solubility, swelling power, and transparency, and the stability of freeze–thaw improved with the increase in DS. Therefore, it can be concluded that acetylation has a certain positive effect on the solubility and freeze–thaw stability of starch, which can overcome the limitation of native starch in ready-to-eat and frozen foods through acetylation modification, thus improving the application value.

## Figures and Tables

**Figure 1 foods-12-02462-f001:**
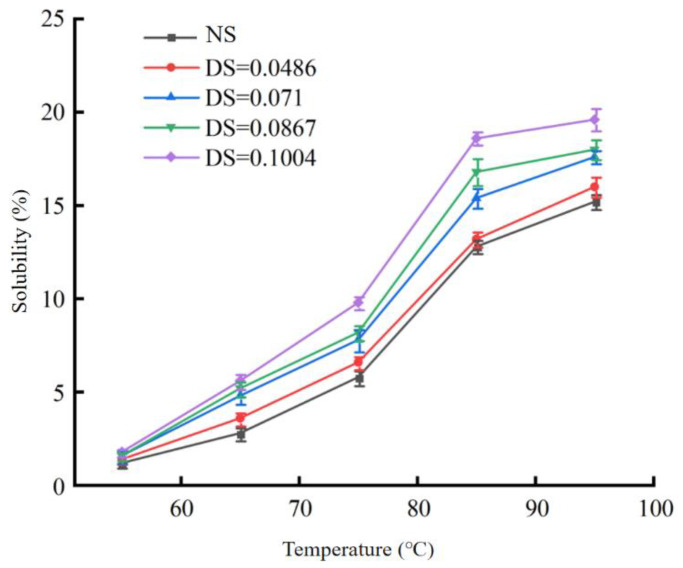
Solubility of acetylated starch. Note: NS refers to natural starch, DS refers to starch with different degrees of substitution, the same below.

**Figure 2 foods-12-02462-f002:**
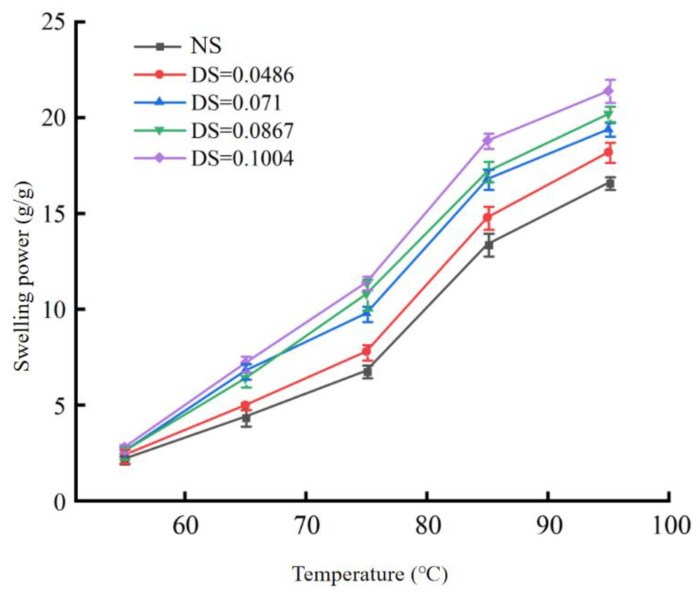
Swelling power of acetylated starch.

**Figure 3 foods-12-02462-f003:**
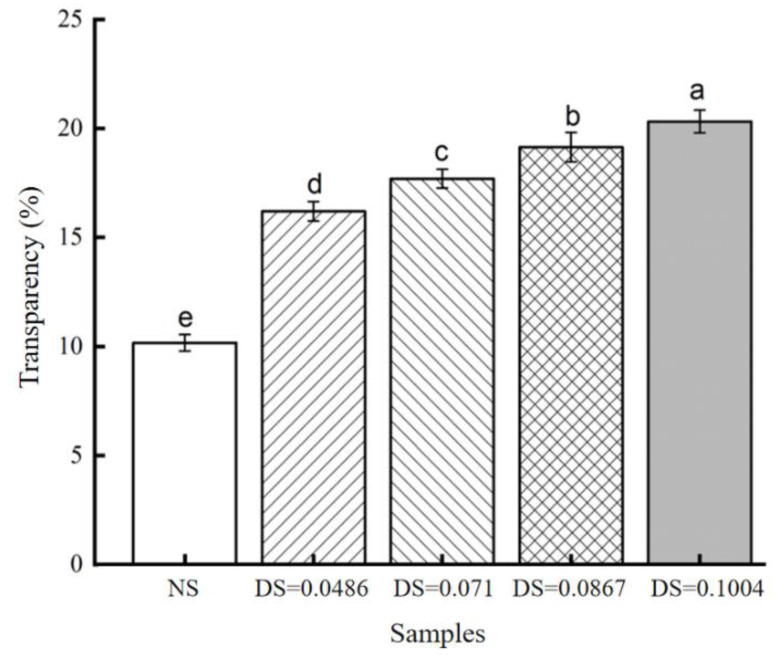
Transparency of acetylated starch. Note: different letters in the picture indicate significant difference (*p* < 0.05).

**Figure 4 foods-12-02462-f004:**
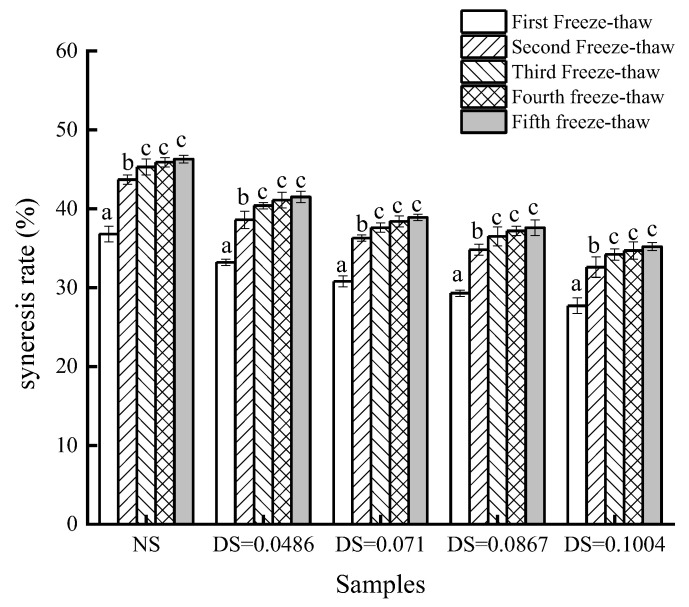
Water syneresis rate of acetylated starch. Note: different letters in the picture indicate significant difference (*p* < 0.05).

**Figure 5 foods-12-02462-f005:**
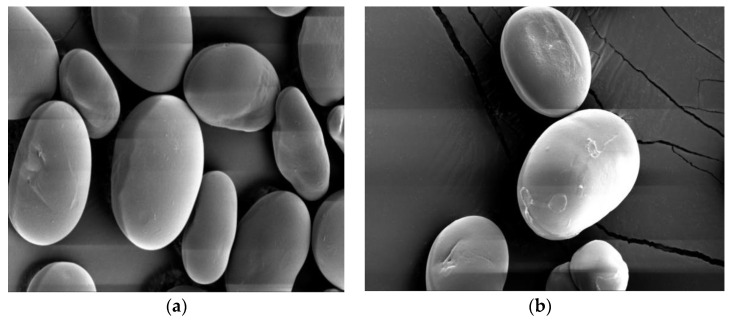
SEM of acetylated chickpea starch with different degrees of substitution ((**a**) NS, (**b**) DS = 0.0468, (**c**) DS = 0.0710, (**d**) DS = 0.0867, and (**e**) DS = 0.1004).

**Figure 6 foods-12-02462-f006:**
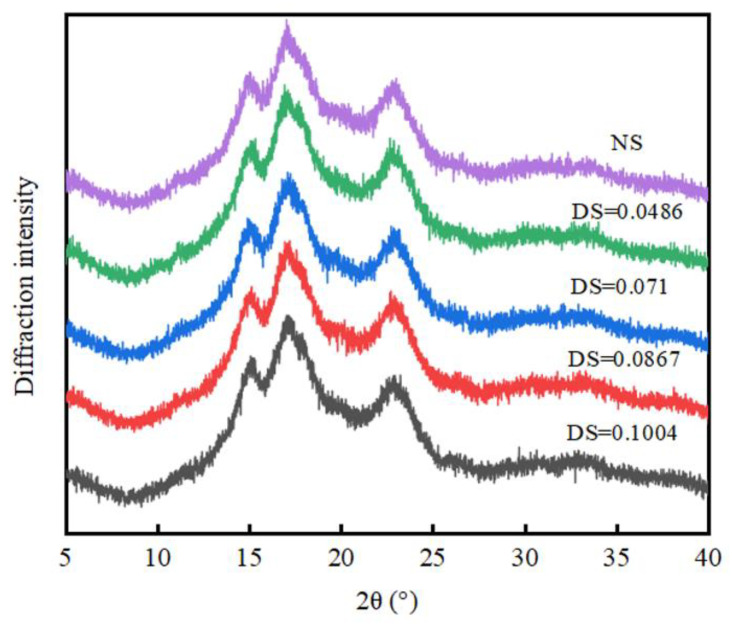
X-ray diffraction pattern of acetylated starch.

**Figure 7 foods-12-02462-f007:**
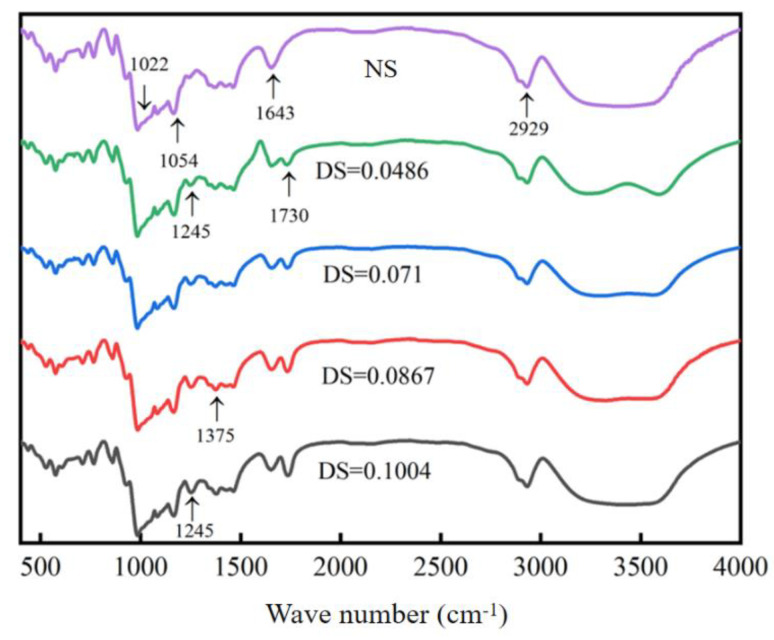
FTIR Spectra of Acetylated Starch.

**Figure 8 foods-12-02462-f008:**
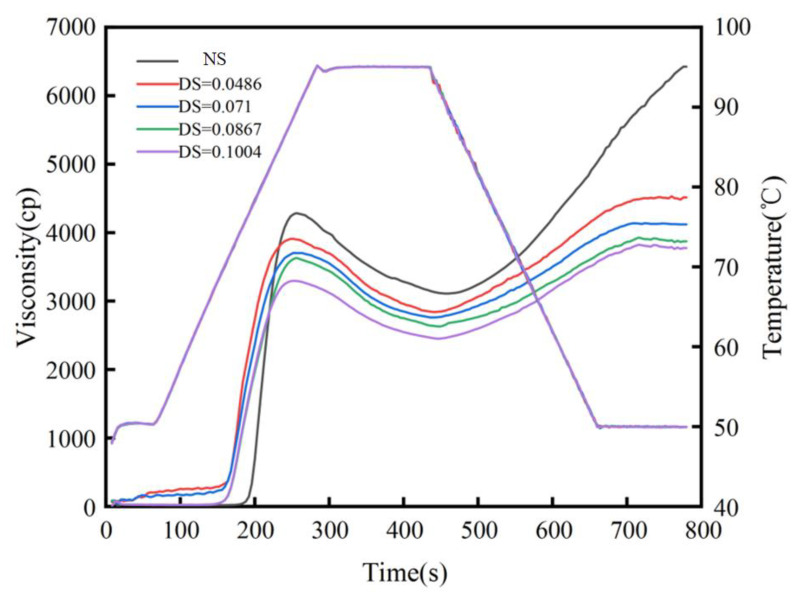
Paste viscosity characteristic curves of native starch and acetylated starch.

**Table 1 foods-12-02462-t001:** Acetyl content and substitution degree of acetylated chickpea starch.

Acetic Anhydride Dosage/%	Acetyl Group Content/%	Degree of Substitution
4	1.29 ± 0.02 ^a^	0.0486 ± 0.002 ^a^
6	1.89 ± 0.03 ^b^	0.071 ± 0.003 ^b^
8	2.30 ± 0.03 ^c^	0.0867 ± 0.001 ^c^
10	2.60 ± 0.02 ^d^	0.1004 ± 0.001 ^d^

Note: different letters in the same column indicate significant difference (*p* < 0.05).

**Table 2 foods-12-02462-t002:** Relative crystallinity of acetylated starch.

Samples	NS	DS = 0.0486	DS = 0.071	DS = 0.0867	DS = 0.1004
Relative crystallinity (%)	31.04	30.89	30.47	30.28	30.07

**Table 3 foods-12-02462-t003:** Paste viscosity parameters of raw starch and acetylated starch.

Scheme	Peak Viscosity (cP)	Trough Viscosity (cP)	Breakdown Viscosity (cP)	Final Viscosity (cP)	Setback Viscosity (cP)	Gelatinization Temperature (°C)
NS	4284 ± 31 ^a^	3110 ± 13.6 ^a^	1174 ± 36.7 ^a^	6424 ± 62 ^a^	3314 ± 57.4 ^a^	75.05 ± 0.4 ^a^
DS = 0.0486	3909 ± 173 ^b^	2841 ± 55 ^b^	1068 ± 171 ^b^	4515 ± 50.7 ^b^	1674 ± 54.2 ^b^	68.65 ± 1.0 ^b^
DS = 0.071	3704 ± 92 ^bc^	2761 ± 52.1 ^bc^	943 ± 38.7 ^c^	4119 ± 13.9 ^bc^	1358 ± 40.8 ^c^	62.53 ± 11 ^b^
DS = 0.0867	3636 ± 48.8 ^b^	2672 ± 45.1 ^bc^	964 ± 40.8 ^b^	3873 ± 70.9 ^cd^	1201 ± 79.9 ^cd^	58.50 ± 0.9 ^bc^
DS = 0.1004	3295 ± 117 ^c^	2450 ± 145 ^c^	845 ± 54 ^cd^	3775 ± 13.3 ^d^	1325 ± 150.5 ^e^	50.25 ± 0.4 ^d^

Note: different letters indicate significant difference (*p* < 0.05).

**Table 4 foods-12-02462-t004:** Gel texture parameters of acetylated chickpea starch.

Species	Hardness/g	Cohesion/g	Elasticity/mm	Adhesion/g·s	Chewiness/mj
NS	29.79 ± 0.58 ^a^	0.76 ± 0.01 ^a^	5.62 ± 0.15 ^a^	19.88 ± 0.4 ^a^	102 ± 5.03 ^a^
DS = 0.0486	15.08 ± 0.63 ^b^	0.75 ± 0.01 ^a^	5.38 ± 0.19 ^a^	10.72 ± 0.23 ^b^	42.4 ± 2.44 ^b^
DS = 0.071	13.15 ± 0.73 ^c^	0.74 ± 0.04 ^a^	5.01 ± 0.23 ^bc^	7.85± 0.27 ^c^	28.2 ± 3.22 ^c^
DS = 0.0867	11.41 ± 0.55 ^d^	0.59 ± 0.03 ^b^	4.54 ± 0.47 ^cd^	6.05 ± 0.49 ^d^	25.6 ± 2.5 ^c^
DS = 0.1004	10.14 ± 0.4 ^e^	0.54 ± 0.03 ^b^	4.12 ± 0.34 ^d^	3.55 ± 0.44 ^e^	22.8 ± 3.27 ^c^

Note: different letters indicate significant difference (*p* < 0.05).

## Data Availability

Data is contained within the article.

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
