# Peer review of "The Effect of Acetylation on the Physicochemical Properties of Chickpea Starch"

_foods, 2023, doi:10.3390/foods12132462_

Round 1

Reviewer 1 Report

The article is about characterization of acetylated chickpea starch, although this starch modification was already studied in chickpea starch, new analysis were done with specific instruments to improve the published data.

However, the author must review another published articles about chikpea starch as:

Effect of acetyl esterification on physicochemical properties of chick pea (Cicer arietinum L.) starch. Dev Kumar Yadavcorresponding author and Prakash Eknatharao Patki. J Food Sci Technol. 2015 Jul; 52(7): 4176–4185.

Is important to do a further article review about chikpea starch modification by acetylation to include in the duscussion section.

In other hand the article doesn´t show experimental design employed in methods section.

Author Response

Point 1:the author must review another published articles about chikpea starch as: Effect of acetyl esterification on physicochemical properties of chick pea (Cicer arietinum L.) starch. Dev Kumar Yadav and Prakash Eknatharao Patki. J Food Sci Technol. 2015 Jul; 52(7): 4176–4185.

Response 1: we have review the article, thank you.

Point 2: Is important to do a further article review about chikpea starch modification by acetylation to include in the duscussion section.

Response 2: We have re-written this part according to the Reviewer’s suggestion, and analyze the effect of acetylation starch functionality.

Point 3: In other hand the article doesn´t show experimental design employed in methods section.

Response 3: We have made correction according to the Reviewer’s comments, methods have been summarized.

Special thank you for your advice to us.

Reviewer 2 Report

The manuscript presents simple experiment regarding acetylation of chickpea starch. The novelty of the research is not stead, the introduction is quite vague. The link with previous research on acetylation of starch is not shown in introduction nor in the discussion of the resuts. The discussion is speculative and the results of should should be confronted with the literature data. Conclusions are mostly properly drawn.

Detailed comments:

Our country - style

Carboxymethylation of starch is not used for production of food grade starch derivatives

Is viscosity and texture is good, and has acertain degree of stability – this needs to be elaborated.

Starch isolation procedure should be provided in detail or employed method cited.

Reactant – reagent?

Please provide SEM acceleration voltage

Formula 5 is not clear

Where the samples for XRD conditioned prior the measurement?

Determine TPA test speeds, now reader can only assume, provide load cell

Table 1 – degree of substitution is not in % (by definition), moreover the DS vlue fo the first sample is too high

Figure 3 swelling power should be in g/g

Figure 6 the angle range in difractogram is to large, it should as stated in methods section to improve clarity

Valley and collapse viscosity, regenerated value? Please use proper scientific terms used in literature

Table 4 please check units in TPA there are not correct

The last paragraph of conclusions in not supported by the results

The manuscript can be understood in most parts, but the style is not correct in several paragraphs .

Author Response

Point 1: Is viscosity and texture is good, and has acertain degree of stability – this needs to be elaborated.

Response 1: Considering the Reviewer’s suggestion, we have explained this in detail.

Point 2: Starch isolation procedure should be provided in detail or employed method cited.

Response 2: Starch extraction methods have been supplemented in this article.

The chickpeas were ground and passed through 80 mesh, then the chickpeas powder 5 g and NaOH solution (0.3% concentration) were mixed according to the ratio of material to liquid 1:4. Under the condition of 40 ℃, it was stirred at constant temperature for 240 min in a magnetic stirring water bath, and centrifuged at 4000 r/min after stirring. After centrifugation, scrape off the top layer of black protein solution, wash the bottom layer of starch with water, centrifuge until the starch is pure white, dry in a 40 ℃ drying box for 24 h, then crush through 80 mesh sieve.

Point 3: Reactant – reagent?

Response 3: We are very sorry for our incorrect writing, corrections have been made in the text.

Point 4: Please provide SEM acceleration voltage

Response 4: The acceleration voltage of SEM is 5 KV.

Point 5: Formula 5 is not clear

Response 5: Figure 5 has been redrawn.

Point 6: Where the samples for XRD conditioned prior the measurement?

Response 6: The sample is dried and pressed.

Point 7: Determine TPA test speeds, now reader can only assume, provide load cell

Response 7: The pre-test speed was 2 mm/s, the mid-test speed was 5 mm/s, and the post-test speed was 5 mm/s.

Point 8: Table 1 – degree of substitution is not in % (by definition), moreover the DS vlue fo the first sample is too high

Figure 3 swelling power should be in g/g

Figure 6 the angle range in difractogram is to large, it should as stated in methods section to improve clarity

Response 8: The substitution units in Table 1 have been corrected. The first sample's DS value was too high due to a writing error and has been corrected from 0.486 to 0.0486.thank you.

Fig. 3 the swelling power has been corrected to g/g

Fig. 6 the diffraction pattern has been redrawn

Point 9: Valley and collapse viscosity, regenerated value? Please use proper scientific terms used in literature

Response 9: Have been described in appropriate scientific terms. Valley Viscosity changed to Trough Viscosity, Collapse value changed to Breakdown Viscosity, and The regenerated value changed to Setback Viscosity.

Point 10: Table 4 please check units in TPA there are not correct

Response 10:Table 4 has been checked. The unit of Cohesion is changed from N to g, and the unit of Adhesion is changed from N to g·s.

Point 11: The last paragraph of conclusions in not supported by the results

Response 11: We have made correction according to the Reviewer’s comments.

Special thank you for your advice to us.

Reviewer 3 Report

The manuscript deals with the effect of acetylation on the physicochemical properties of chickpea starch.

The English language must be revised.

Please separate values from units, “4 ºC” not “4ºC”.

Abstract

Please present your main results.

Introduction

This section is short and must be improved.

Materials and methods

“The scanning electron microscope (Apero S, Lecht instruments and Equipment Co., Ltd. Shenzhen, China) was used to observe the surface morphology of chickpea starch…”??voltage used??

A statistical section is missing.

Results and discussion

This section must be revised. Some results must present a suitable statistical analysis. Moreover, please revise the discussion in accordance.

Conclusion

Please do not repeat your results and focus on your main conclusions.

References

Please format the scientific names in italic.

The English language must be revised.

Author Response

Point 1: Please separate values from units, “4 ºC” not “4ºC”. 

Response 1: We have made corrections in the text, 4℃ changed to 4 ℃.

Point 2: Abstract

Please present your main results.

Response 2: We have rewritten the Abstract according to the Reviewer’s comments and have presented only the main results.

Point 3: Introduction

This section is short and must be improved. 

Response 3: We have rewritten the Introduction according to the Reviewer’s comments.

Point 4: Materials and methods

“The scanning electron microscope (Apero S, Lecht instruments and Equipment Co., Ltd. Shenzhen, China) was used to observe the surface morphology of chickpea starch…”??voltage used??

A statistical section is missing. 

Response 4: The acceleration voltage of SEM is 5 KV. The Statistics section has been supplemented in this article.

Point 5: Results and discussion

This section must be revised. Some results must present a suitable statistical analysis. Moreover, please revise the discussion in accordance. 

Response 5: The Results and discussion has been modified according to the Reviewer’s comments.

Point 6: Conclusion

Please do not repeat your results and focus on your main conclusions. 

Response 6: The conclusion has been rewritten to focus only on the main conclusion according to the Reviewer’s comments.

Point 7: Please format the scientific names in italic. 

Response 7: We have used italics for the scientific names. For example, Chickpea changed to Chickpea.

Special thank you for your advice to us.

Round 2

Reviewer 2 Report

The manuscript has been improved, however the TPA units need to be rechecked.

Author Response

Point 1:

The manuscript has been improved, however the TPA units need to be rechecked.

Response 1: we have rechecked the units and corrected it , thank you.

Reviewer 3 Report

Please replace “5 KV” by “5 kV”.

Minor corrections are needed.

Author Response

Point 1 Please replace “5 KV” by “5 kV”.

Response 1: we have replace "5KV" by "5kV", thank you.

Point 2  Minor corrections are needed.

Response 2: We checked the manuscript, and the changes were marked in red. thank you